

# A social recommendation model based on adaptive residual graph convolution networks

Rui Chen[1], Kangning Pang[1], Qingfang Liu[2], Lei Zhang[3], Hao Wu[4], Cundong Tang[4] and Pu Li[1]

[1] College of Software Engineering, Zhengzhou University of Light Industry, ZhengZhou, Henan, China
[2] Information Center, Shijiazhuang Posts and Telecommunications Technical College, Shijiazhuang, Hebei, China
[3] School of Mathematics and Information Technology, Yuncheng University, Yuncheng, Shanxi, China
[4] School of Information Science and Technology, Northwest University, Xi'an, Shanxi, China

Corresponding author
Qingfang Liu, liuqf@sjzpc.edu.cn

## ABSTRACT

Incorporating social information in the recommendation algorithm based on graph neural network (GNN) alleviates the data sparsity and cold-start problems to a certain extent, and effectively improves the recommendation performance of the model. However, there are still shortcomings in the existing studies: on the one hand, the potential effect of noise in the raw data is ignored; on the other hand, only relying on the single interaction information between the user and the item and failing to make full use of the rich multi-aided information. These factors lead to an unsatisfactory learning effect of the model. To address the above problems, we propose a social recommendation model based on adaptive residual graph convolutional networks (SocialGCNRI). Specifically, we use the idea of fast Fourier transform (FFT), a filtering algorithm in the field of signal processing, to attenuate the raw data noise in the frequency domain, followed by utilizing the user-social relations, item-association relations, and user-item-interaction relations to form a heterogeneous graph to supplement the model information, and finally using a graph convolution algorithm with an adaptive residual graph to improve the expressive power of the model. Extensive experiments on two real datasets show that SocialGCNRI outperforms state-of-the-art social recommendation methods on a variety of common evaluation metrics.

## INTRODUCTION

Recommender systems effectively mitigate the information overload problem and greatly improve users' web experience (*Aljunid & Huchaiah, 2021*; *Koren, Rendle & Bell, 2021*; *Wang et al., 2019b*). However, the recommendation algorithms themselves face data noise and cold-start problems that are difficult to alleviate (*Chen et al., 2019*; *Ma et al., 2008*). With the rapid development of information technology and the Internet, social software such as Wechat, Amazon, Facebook, *etc.* are growing, which greatly enriches the social

information among users, and people gradually realize that they can be used to optimize the performance of recommendation systems. In essence, social behavior is an inherent part of human activities, and the viewpoints of friends often have a significant impact on users' decisions (*Lewis, Gonzalez & Kaufman, 2012*; *Iyengar, Han & Gupta, 2009*). As a result, social recommendation algorithms have emerged, which take social information among users as an important supplement to enrich user information from multiple dimensions, thus significantly improving the performance of recommender systems and attracting widespread attention (*Zhao et al., 2017*; *Yu et al., 2019*).

In recent years, graph neural networks (GNNs) have achieved remarkable success in node classification and link prediction tasks. Benefiting from its ability to effectively encode graph-structured data, GNNs are able to dig deeper into the interrelationships between nodes (such as users and items) as well as the overall structural features of the graph, a property that has led to the increasing application of graph neural network technology in the field of recommender systems (*Kipf & Welling, 2016*). In the context of social recommendation, the social network topology among users and the interaction records between users and items can be mapped into graph data form. Therefore, social recommendation algorithms relying on GNN architecture have gradually become the research hotspot and mainstream trend in this field. This type of methods dramatically improves the accuracy and generalization ability of recommendation systems by accurately identifying the interconnections between nodes and the deep structural information in the graph. For example, GraphRec (*Fan et al., 2019*) learns embedding functions in user social graphs and user-item interaction graphs using GNN, which greatly enriches user and item information. Some other researchers combine implicit friends with similar preferences as supplemental information with explicit social information to form a heterogeneous social network, which expresses user preferences more accurately. SocialLGN (*Liao et al., 2022*) designs a component to fuse the rich user information in a heterogeneous social network, which improves the recommendation accuracy. SlightGCN (*Jiang et al., 2022*) designs different perspectives in the heterogeneous network as auxiliary information to improve the embedding quality and enhance the model accuracy. These studies have significantly improved the recommendation accuracy by incorporating social information in different ways, but there are still some challenges: (1) The noise problem in the initial information greatly affects the recommendation performance; (2) the auxiliary information in the existing data has not been sufficiently mined; (3) the current model architecture makes it difficult to mine the higher-order connections between nodes.

To address these problems, we propose a social recommendation model based on adaptive residual graph convolutional networks (SocialGCNRI). The model constructs sufficient input information for the model through multiple auxiliary information, and uses an adaptive residual graph convolution algorithm to better learn user representation information, which greatly improves the recommendation performance. Specifically, we modify the fast Fourier transform (FFT) algorithm in the signal processing domain so that it can be applied to the recommendation domain to alleviate the noise problem in the initial information. Then, the original information is supplemented by mining additional user social relationships and item associations through the interaction information

between users and items. Finally, we design an adaptive residual graph convolution algorithm to fully explore the deep connections between the nodes in the graph, so that the model can be more fully expressed and the accuracy of the model can be substantially improved.

The main contributions of this article are as follows:

(1) We propose a new social recommendation model based on GNN. The model targets noisy information in the data and complements the information sources by mining the user item interaction graph.

(2) We apply the idea of FFT in the signal processing field to the recommendation field, through the combination of FFT and filter to deal with the noise information in the original data, to reduce the impact of the noise problem on the accuracy of the recommendation.

(3) We design an adaptive residual graph convolution algorithms. In the process of graph convolution, the similarity between the current embedding layer and the initial embedding layer is used to adaptively supplement the initial embedding information, effectively delaying the occurrence of graph smoothing phenomenon, mining the deep connection between the nodes, and providing higher-quality recommendations for the target users.

(4) SocialGCNRI is applied to two real datasets to compare and validate with different baseline methods, and the experimental results demonstrate the superiority of the model on three metrics, Recall, Precision, and NDCG, and further ablation experiments validate the effectiveness of the model components.

The remainder of the article is organized as follows. "Related Work" presents the related work in this article. "Proposed Model" describes the design details of the proposed model. "Experiments" reports the experimental findings. "Conclusion" summarizes the article.

## RELATED WORK

In this section, we first introduce denoising algorithms among recommender systems, and then provide a detailed overview of related work in social recommendation.

### Cold-start recommendation models

In the development of recommendation algorithms, the cold-start problem has always been one of the directions that researchers focus on. When the number of user-item interactions is too small, the recommender system will reach a cold start state when it cannot effectively make personalized recommendations for the user. For this phenomenon, Generative Adversarial Recommendation (GAR) (*Chen et al., 2022*) uses generative adversarial modeling to generate user interaction data, and distinguishes between generated and real data in the form of fine-tuning during recommendation ranking, thus mitigating the cold-start problem. In difference, *Huang et al. (2023)* think that this model is not effective in mitigating the variability between hot and cold items, and they propose the Aligning Distillation (ALDI) model, which reduces these differences by tailoring the rating alignment and identifying alignment losses, and using a weighting

structure to ensure that the model learns accurately with respect to the information. For the cold-start problem in large-scale online recommendation, *Huang et al. (2025)* proposed the large language model (LLM) simulator framework, which reduces the number of candidate recommendation users by coupling filters, and uses a LLM to simulate the interaction of cold-start users, which not only successfully alleviates the cold-start problem, but also reduces the complexity of the model. On this background, *Zhang et al. (2025)* further analyzes the effectiveness of LLM for the cold-start problem in large-scale online recommendation, which aims to provide new insights for the related workers. The above methods effectively mitigate the cold-start problem in recommendation through generative models or LLM, but generative models tend to make the recommendation of the model too polarized, while LLM have the problem of excessive complexity. In contrast, SocialGCNRI builds a heterogeneous graph to enrich the user-item interaction information by means of user social information, which not only can effectively alleviate the occurrence of the cold start phenomenon, but also ensures that only the user-item and user social matrices will be used in the process of building the heterogeneous graph, so that the complexity of the model can be ensured to be within the controllable range.

## De-biasing algorithms on recommendations

GNN greatly improve the accuracy and robustness of recommendation models by modeling users and items, but may amplify item bias during graph aggregation, which makes the model's recommendations result in a long-tail phenomenon. Noting this kind of problem, *Chen et al. (2020a)* defines seven types of bias in recommendation, and detailed the role of different loss functions for removing bias. In contrast, *Zhou et al. (2023)* thinks that the way of modifying the loss function does not completely solve the problem of bias in the model, and they propose a novel graph aggregator, the adaptive popularity debiasing aggregator (APDA), to learn the weights of each edge and the aggregation popularity bias, and use the weight scaling mechanism and residual connection to eliminate the bias. We also believe that the use of aggregators is one of the ways to address the problem of bias, and in SocialGCNRI, we propose an adaptive residual map convolutional network to remove bias, and use a multilayer perceptron (MLP)-based aggregator to further mitigate the emergence of long-tail phenomenon in the model.

## De-noising algorithm on recommendation

The impact of uncertainties such as mis-touchs, false comments, sample bias, and so on during user interactions makes it inevitable that user log files will contain noisy data. It is worth noting that noise in recommendation algorithms is very similar to the Sybil attack, which is an attack in which a malicious user gains undue influence by creating a large number of false identities and using those identities to participate in system operations, both of which cause the model to react adversely by injecting false information. The presence of noisy data often leads to large fluctuations in model performance, thus reducing the accuracy of recommendations. To address this problem, more and more scholars have began to design denoising algorithms to mitigate the negative impact of

noisy data on model performance (*Wang et al., 2021*; *Joorabloo, Jalili & Ren, 2022*). In order to extract effective features in implicit interactions, the Stacked Discriminative Denoising Auto-Encoder based Recommender System (SDDRS) (*Wang et al., 2019a*) designs a stacked discriminative denoising auto-encoder to remove noisy data from interactions, effectively combining implicit interaction information with rating information. In the multi-modal recommendation task, collaborative denoised graph contrastive learning (CDGCL) (*Xu et al., 2024*) designs a multi-strategy denoising module to filter out irrelevant interaction information as a means of accurately and precisely reflecting the user's true interests. In summary, different ways of denoising the raw data can effectively improve the accuracy of recommendation.

## Social recommendation with graph neural networks

In traditional recommender systems, the interaction data between users and items are often sparse, and this characteristic limits the accuracy of the recommendation results. To alleviate this phenomenon, researchers have proposed social recommendation algorithms based on the social influence theory, which take the social information between users as supplemental information to alleviate the limitations of traditional recommendation algorithms. Most of the early social recommendation models used matrix factorization (MF) techniques and fused social information into the recommendation model through various strategies with a view to improving the recommendation performance (*Tang et al., 2016*; *Zhang et al., 2018*; *Yang et al., 2016*; *Parvin et al., 2019*). For example, SocialMF (*Jamali & Ester, 2010*) cleverly incorporates social influences into the recommendation model by considering user feature vectors as weighted combinations of their friends' feature vectors. On the other hand, Social Bayesian Personalized Ranking (SBPR) (*Zhao, McAuley & King, 2014*) constructs a Bayesian framework aimed at filtering out more similar friends for the user, thus skillfully integrating social information into the collaborative filtering process. Although social recommendation models based on matrix factorization overcome some of the shortcomings of traditional recommendation to a certain extent, they are still deficient in capturing complex social network structures and utilizing node attribute information.

In recent years, more and more researchers have used GNN techniques to solve problems among social recommendations (*Yu et al., 2018*; *Zhao et al., 2019*). The GNN-based social recommendation learns target node embedding by aggregating neighbor information in the user-item interaction graph and social network graph (*Salamat, Luo & Jafari, 2021*). GraphRec (*Fan et al., 2019*) combines social information with GNN for the first time, and utilizes GNN technology to enrich the attribute information of user nodes by analyzing their first-order neighbor relationships. DiffNet (*Wu et al., 2019a*) designs a hierarchical propagation mechanism to realize the dynamic propagation of social influence in social networks, and as a result captures a multilevel representation of user information in different graphs. The light graph convolution network for social recommendation (SocialLGN) (*Liao et al., 2022*) employs light graph convolution operation to extract attribute information about user nodes in different graphs. In addition, the model is designed with a specialized fusion component for

combining the extracted social information with other features of the user. The robust social recommendation based on contrastive Learning and dual-stage graph neural network (CLDS) (*Ma et al., 2024*) utilizes comparative learning techniques to extract deep features of users and items, and capture the influences in social networks to improve the performance of graph neural networks in social recommendation tasks.

The above methods utilizes a GNN approach to capture users' preferences for different items, which significantly improves the model performance, but cannot effectively capture the higher-order connections between nodes. Different from them, we design a graph convolution network based on adaptive residuals, by adaptively adjusting the weight of the user's initial embedding to continuously update the user's attribute information, we can more accurately infer the user's preference for different items and improve the model accuracy.

# PROPOSED MODEL

In this section, we first provide an overview of the general framework of the SocialGCNRI, and then describe each component of the model in detail.

## Overall framework of the SocialGCNRI

The overall framework of SocialGCNI is shown in Fig. 1, which mainly consists of embedding layer, filter layer, residual graph convolutional layer and prediction layer. When the user-item interaction graph and the user social graph are inputted into the model, it goes through a series of processing steps to finally generate a set of recommended items for the target user.

Taking the target user as an example, SocialGCNRI first generates the user's ID embedding through the embedding layer. This embedding is used as an input to the filtering layer, where the FFT is used to filter the noisy data in the user interaction sequence, and the dropout operation is used afterwards to prevent over-fitting. Subsequently, in the residual graph convolution layer, an adaptive residual graph convolution operation is used in the user interaction graph and the user social graph to further extract the user's preference representations, and in the prediction layer the predicted score of user $u_a$ for item $i_b$ is computed.

## Embedding layer

At the embedding layer, we project the high-dimensional one-hot representations of the user and the project into a fixed-length low-dimensional dense representation: $E_u \in \mathbb{R}^{n \times d}$ and $E_i \in \mathbb{R}^{m \times d}$, respectively.

$$E_u = [e_{u_1}, e_{u_2}, ..., e_{u_n}], E_i = [e_{i_1}, e_{i_2}, ..., e_{i_m}] \tag{1}$$

where $E_u$ is a matrix consisting of n user elements and d is a hyper-parameter denoting the length of a single user embedding vector. $E_i$ is a matrix consisting of m item elements and d is a hyper-parameter denoting the length of a single item embedding vector. Specifically, given a one-hot representation of the target user $u_a$, the embedding layer performs indexing and outputs the transpose of the a-th row of the user embedding matrix

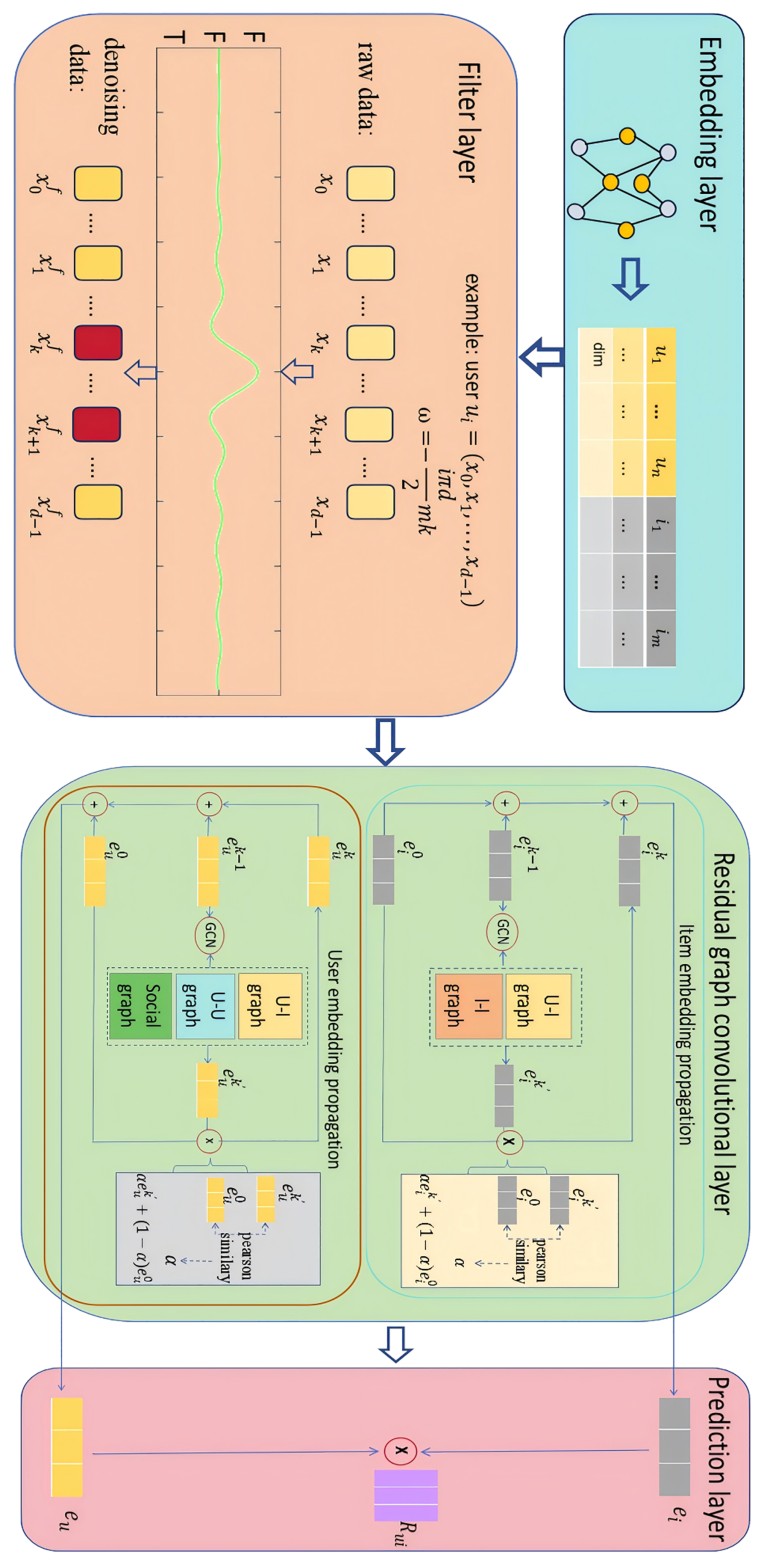

**Figure 1 Overall framework of the SocialGCNRI.**

$E_u$. Similarly, given a one-hot representation of the target item $i_b$, the embedding layer performs indexing and outputs the transpose of the b-th row of the item embedding matrix $E_i$.

## Filter layer

Social recommendation algorithms based on graph convolutional networks (GCN) predict user preferences by analyzing the user's interaction history and the influence of social friends. However, the user's interaction history generates noisy data due to a series of misoperations such as user's mis-touch, mis-submission, and mis-entry, which makes the model unable to accurately predict the user's preference. To alleviate the impact of this noisy data on recommendation accuracy, we employ the FFT in the field of signal processing to eliminate the noisy data (*Zhou et al., 2022*). Specifically, we apply the FFT after the embedding layer to transform the vectors into the frequency domain, perform noise attenuation through a filter, then use the inverse FFT to transform the noise-attenuated data back into vectors again, and finally use the Dropout operation to prevent overfitting. We will describe the operation of the filtering layer from the user's point of view, and for the item embedding, we will perform the same operation to obtain the filtered embedding representation.

### Fast Fourier transform

Given the user's ID embedding matrix $E_u \in \mathbb{R}^{n \times d}$, we will perform an FFT operation along the item dimension to transform each user's interaction vector into the frequency domain:

$$E^f = F(E_u) \in \mathbb{C}^{n \times d} \tag{2}$$

where $E^f$ is a complex tensor denoting the spectrum of $E_u$. $F(\cdot)$ denotes a one-dimensional FFT operation, and for user $u_a = (x_0, x_1, \ldots, x_{d-1})$, the FFT is formulated as:

$$F(\cdot) = C_x = \sum_{m=0}^{d-1} x_m \omega_{km}^m \ldots\ldots 0 \le k \le d - 1 \tag{3}$$

where $C_x$ denotes the representation of user $u_a$ in the frequency domain $\omega_{km}^m$. For each k, the FFT generates a new representation $C_x$. $\omega_{km}^m = -i\frac{2\pi}{d}mk$ denotes the rotation factor generated by the Cooley-Tukey algorithm (*Shirbhate, Panse & Ralekar, 2015*), and $i$ denotes the imaginary numbers. User-item interaction signals represented in the frequency domain can reveal features that are difficult to observe in the time domain, which facilitates the use of filters to suppress noise:

$$\bar{E}^f = W \cdot E^f \tag{4}$$

where $W \in \mathbb{C}^{n \times d}$ is a learnable filter. It can be optimized by stochastic gradient descent (SGD) to adaptively attenuate noisy data in the spectral domain.

### Inverse fast Fourier transform

The inverse FFT is used to reconvert the complex tensor processed in the frequency domain into a real tensor, and to update the embedding representation:

$$\widehat{E}_u = F^{-1}(\bar{E}^f) \qquad (5)$$

where $F^{-1}(\cdot)$ denotes the inverse FFT in one dimension. The formula for performing the inverse FFT on the user $u_a$ spectral domain representation $C_x$ is expressed as:

$$F^{-1}(\cdot) = \frac{1}{d}\sum_{n=0}^{d-1} C_n \cdot (-\omega_m^{km}). \qquad (6)$$

Through the combined operation of FFT and filter, the filter layer of SocialGCNRI can effectively reduce the noisy data in the original user-item interactions, so as to obtain embedded representations that better fit the user's preferences. Considering the problems of gradient vanishing and training instability, we use dropout operation after inverse FFT:

$$E_u^{(0)} = E_u + dropout(\widehat{E}_u). \qquad (7)$$

It is noteworthy that the construction of the FFT component is based on the MLP architecture, which has a more simplified architecture than other comparable models (such as CLEA (*Qin, Wang & Li, 2021*), BERT4Rec (*Sun et al., 2019*), and SRGNN (*Wu et al., 2019b*)), and is simple and efficient in real-world model training.

## Residual graph convolutional layer

The propagation mechanism in GNN mines the relationships of nodes in the graph through information transfer between nodes, and uses this as a basis for accurately capturing user preferences. Most of the current recommendation algorithms based on GCN use LightGCN to accomplish the information propagation between nodes in the graph, and this graph convolutional algorithm removes linear transformations and nonlinear activation operations, which greatly reduces the computational complexity and simplifies the traditional graph convolutional network architecture (*He et al., 2020*). However, the simple propagation process makes the convolutional layer fall into the smoothing problem prematurely and fails to extract the higher-order relationships among nodes effectively. Inspired by GCNII (*Chen et al., 2020b*), in order to alleviate the problem of node representations falling into the smoothing problem as they are diluted by the neighbor information during the propagation process, we design an adaptive residual graph convolutional network performed in heterogeneous graphs to mine the node information in the graphs more effectively.

### *User-item heterogeneous graph construction*

In social recommendation situations, user nodes are influenced by friends and historical interaction items, while item nodes are influenced by similar items and historical interaction users. However, the information in the original user-item interaction matrix tends to be sparse. Inspired by SLightGCN (*Jiang et al., 2022*), we use meta-path $\phi_{user}(user\text{-}item\text{-}user)$ to generate user friend information, generate item association relations *via* meta-path $\phi_{item}(item\text{-}user\text{-}item)$, and integrate them with user-item

interactions in the form of edges into a heterogeneous graph $G_1$, based on which the generated matrix is defined as:

$$A = \begin{pmatrix} S & R \\ R^T & S^T \end{pmatrix} \tag{8}$$

where $S = RR^T$ is the user friend relationship generated through meta-path $\phi_{user}$. $S^T = R^T R$ is the item association relationship generated through meta-path $\phi_{item}$.

The construction of heterogeneous graph $A$ is mainly based on the friendship matrix $S$ between users and the association matrix $S^T$ between items, the smallest unit of the composition matrix is $R \in \mathbb{R}^{n \times m}$ then the time of composing S is $n^2$, and the time complexity of composing $S^T$ is $m^2$, then the time complexity of heterogeneous graph A is about equal to $O(n^2)$. In the face of the expansion of large-scale graphs, the heterogeneous graphs are able to integrate the characteristics of different data types in the graphs and more comprehensively portray the complex relationships in the recommended scenarios, but the graphs construction time complexity still needs to be further compressed.

### Information propagation between nodes

For user embedding, we generate user representation $q_u$ through user item isomorphism graph $G_1$, and user representation $G_r$ through user social relationship graph $p_u$, and then fuse the user representations under different perspectives to obtain the final embedding vector of the user. In a graph convolutional network with k propagation layers, the formula for the construction of user information in the $l$-th layer are defined as:

$$q_u^{(l)} = \alpha \widehat{A} q_u^{(l-1)} + (1 - \alpha) q_u^{(0)} \tag{9}$$

$$p_u^{(l)} = \alpha \widehat{A}_2 p_u^{(l-1)} + (1 - \alpha) p_u^{(0)} \tag{10}$$

where $q_u^{(l)}$ and $p_u^{(l)}$ denote the user information at $l$-layer obtained under different viewpoints, and $q_u^{(0)} = p_u^{(0)} = E_u^{(0)}$ denotes the embedded information at layer 0 of the user. $\widehat{A} = D^{-\frac{1}{2}} A D^{-\frac{1}{2}}$ is the laplace operator constructed *via* the user-item heterogeneous graph, and D denotes the degree of the matrix A. Similarly, $\widehat{A}_2 = D^{-\frac{1}{2}} A_s D^{-\frac{1}{2}}$ is the laplace operator constructed *via* the user social graph and $A_s$ denotes the user social matrix. $\alpha$ is a learnable tuning parameter, it controls the weight of the initial connection information by calculating the similarity between the current user embedding and the original user embedding through the Pearson's correlation coefficient, which is defined as follows:

$$\alpha = max(pearson(E_u^{(l-1)}, E_u^{(0)}), 0) \tag{11}$$

where 0 is to prevent a negative condition of the adjustment parameter. Through the learning of $\alpha$, the residual graph convolution layer can adaptively adjust the proportion occupied by the original layer, effectively delaying the occurrence of graph smoothing phenomenon. After k propagations, we obtain the embedded representation of the user in each view by accumulating them:

$$q_u = sum(q_u^{(0)}, ..., q_u^{(k-1)})/k, \tag{12}$$

$$p_u = sum(p_u^{(0)}, ..., p_u^{(k-1)})/k. \tag{13}$$

Compared with LightGCN, SicialGCNRI in the time complexity of the extra part of the adaptive residual operation, each layer of the graph convolution operation after the adaptive residual operation of the time complexity of $O(n)$, $l$ layers of convolution under the extra time complexity of $O((l-1)n)$, the increase in time complexity of the same order of magnitude, fully adaptable to the expansion of the operation of the large-scale graph.

Following the strategy in SocialLGN (*Liao et al., 2022*), we use a multi-layer perceptron (MLP) fusion-based component to generate a final embedded representation of the user, which is defined as:

$$\widehat{E}_u = \frac{W_3(tanh(w_1 q_u) \parallel (tanh(w_2 p_u))}{\parallel W_3(tanh(w_1 q_u) \parallel (tanh(w_2 p_u)) \parallel_2} \tag{14}$$

where $tanh$ denotes the nonlinear activation function. $W_1, W_2 \in R^{d \times d}$, $W_3 \in R^{d \times 2d}$ are the trainable weight matrices. For the item representation, we use the user-item isomorphism graph $G_1$ to perform the item embedding propagation operation, and obtain the final item representation by accumulation, which are defined as:

$$E_i^{(l)} = \alpha \widehat{A} E_i^{(l-1)} + (1 - \alpha) E_i^{(0)} \tag{15}$$

$$\widehat{E}_i = sum(E_i^{(0)}, ..., E_i^{(k-1)})/k \tag{16}$$

where $E_i^{(l)}$ denotes the embedded representation of the item obtained in the $l$-th layer, and $E_i^{(l)}$ denotes the final embedded representation of the item obtained through the propagation layer.

## Prediction layer and model optimization

### Prediction layer

With the propagation operation of adaptive residual graph convolution, we can obtain the final embedding vector $\widehat{e}_u$ for user u and the final embedding vector $\widehat{e}_i$ for item i. In this way, we can predict the ranking score $\widehat{y}_{ui}$ of user u for item i by means of the inner product, which is defined as:

$$\widehat{y}_{ui} = \widehat{e}_u^T \widehat{e}_i \tag{17}$$

### Model optimization

We use a Bayesian personalized ranking (BPR) loss function for model optimization, which not only optimizes the user's ranking of items with known preferences, but also takes into account the user's potential preferences for unobserved items, this greatly improves the model's ability to discriminate between similar positive and negative samples. We define the BPR loss to optimize the model parameter $\Phi = \left\{ e_u^{(0)}, e_i^{(0)} | u \in U, i \in I \right\}$, which is defined as:

$$Loss_{bpr} = \frac{1}{|\mathcal{N}|} \sum_{(u,i,j) \in \mathcal{N}} -\sigma(\widehat{y}_{ui} - \widehat{y}_{uj}) + \lambda ||\Phi||_2^2 \tag{18}$$

where $\mathcal{N} \subseteq \{(u, i, j) | (u, i) \in R^p, (u, j) \in R^n\}$ denotes the sampling data of the mini-batch. $R^p$ denotes the positive sampling data observed in the user-item interaction graph, and $R^n$ denotes the negative sampling data not observed in the user-item interaction graph. $\sigma$ denotes the Sigmoid activation function. $||\cdot||_2^2$ denotes the regularization function, and $\lambda$ is the hyper-parameter used to control the regularization of $L_2$.

# EXPERIMENTS

In this section, we apply the SocialGCNRI model to two real datasets for a series of experiments and compare the results with other baseline models, mainly verify the following questions:

(1) The advantages of SocialGCNRI in terms of recommendation performance.
(2) SocialGCNRI effectively mitigates noisy data and cold start problems.
(3) Impact of the FFT algorithm and adaptive residual graph convolutional algorithm on recommendation performance.
(4) Impact of the number of layers of adaptive residual graph convolutional algorithm on recommendation performance.

## Datasets

We employ the LastFM and Ciao datasets, which are commonly used in social recommendation, to evaluate the performance of SocialGCNRI as well as the benchmark model. These datasets vary greatly in number and sparsity, and the specific statistics are shown in Table 1. LastFM is a widely used music recommendation datasets, which includes user music preference networks and social network. The Ciao datasets is derived from the famous online shopping platform Ciao, which contains a large number of product reviews and ratings from consumers around the world. We consider the movie part of the datasets, which includes users' movie preferences and social networks. In the data preprocessing, in order to effectively extract the user-item interaction information, we delete the data nodes in the datasets with less than 20 interactions to constitute the normal datasets, and call the data nodes with less than 20 interactions as the cold-start data nodes, so as to construct the cold-start datasets.

## Evaluation metrics

We use Precision@k, Recall@k, and Normalized Discounted Cumulative Gain (NDCG@k) to evaluate model performance. These metrics are described in detail as:

**Precision@k** calculates the weight of correct recommendations among all recommendations to evaluate the quality of the recommendation results, which is defined as:

$$Precision@k = \frac{TP@k}{TP@k + FP@k} \tag{19}$$

where TP@k denotes the number of recommendation items selected by the user, and FP@k

**Table 1 Statistics of the LastFM and Ciao datasets.**

| Datasets | Number of users | Number of item | Number of user-item interaction | Density of interaction | Number of social connections | Density of connections |
|---|---|---|---|---|---|---|
| LastFM | 1,892 | 17,632 | 92,834 | 0.028% | 25,434 | 0.711% |
| Ciao | 7,375 | 105,114 | 284,086 | 0.037% | 57,544 | 0.016% |

denotes the number of recommendation items not selected. With this evaluation, we can visualize the accuracy of the model.

**Recall@k** measures the proportion of the recommender system that includes items of interest to the user in the first K items it recommends, which is defined as:

$$Recall@k = \frac{R(u)@k \cap T(u)@k}{T(u)@k} \tag{20}$$

where R(u)@k denotes the top K items recommended for the user. T(u)@k denotes the items that are actually of interest to the user as determined by the test datasets. Recall@k is used to calculate the proportion of user-interested items in the first K recommended items among all the items, and a higher value indicates that the recommender system is able to capture the user's interest more effectively.

**NDCG@k** considers the relevance of the recommendation items and their position in the recommendation list, which is defined as:

$$NDCG@k = \frac{DCG@k}{IDCG@k} \tag{21}$$

where DCG@k denotes the weighted accumulation of the top k items in the recommendation list, and IDCG@k denotes the DCG values of the top k items ideally sorted according to relevance from highest to lowest. NDCG@k uses the normalized scores of IDCG@k to DCG@k to assess the overall quality of the top-k items in the recommendation list.

## Baseline models

We use mainstream baseline models under different types as a contrast to evaluate the performance of the SocialGCNRI.

SBPR (*Zhao, McAuley & King, 2014*): This is a Bayesian personalized ranking algorithm based on similar friends in social networks, which is mainly used to solve the problem of data sparsity and single-class collaborative filtering in recommender systems.

DiffNet (*Wu et al., 2019a*): This model uses a graph neural network to model the recursive social diffusion process of each user, capturing the diffusion of influence hidden in higher-order social networks during user embedding.

NGCF (*Wang et al., 2019b*): It extends the idea of collaborative filtering to the field of graph neural networks, and designs a novel propagation mechanism to fuse the higher-order neighbor embedding information of nodes.

LightGCN (*He et al., 2020*): A light recommendation model that removes linear operations and nonlinear activation functions from NGCF, greatly simplifying model complexity.

SocialLGN (*Liao et al., 2022*): A social recommendation model that incorporates social information into the composition of LightGCN. It designs a graph fusion component to compute the embedded representations from different perspectives of the user.

DSL (*Wang, Xia & Huang, 2023*): A social recommendation model for contrastive learning. It designs a cross-view to filter out noisy social influences based on the interaction preferences of different users, thus improving the model accuracy.

CLDS (*Ma et al., 2024*): A recommendation model based on two-stage comparative learning. It uses a graph aggregation algorithm to mitigate the noise in the user-item interaction graph, and compares the user interaction representation with the user social representation to enhance the accuracy and robustness of the recommendation model.

## Experiment setup

Following the strategy described in recent experiments (*Long et al., 2021*), we divide the datasets into training, testing, and validation sets in the ratio of 8:1:1. We use the Python framework to build the SocialGCNRI model, and the model optimizer is Adam. For the hyper-parameters in the model, we set the Dropout to 0.5, the learning rate lr to 0.01, the number of iterations epoch to 10,000 (terminate the training early if the model has no metrics growth for 30 consecutive times), the batch size to 100, the item embedding dimensions d to $\{32, 64, 128, 256\}$, and the number of graph convolution layers to $\{2, 3, 4\}$. To ensure the fairness of the experimental data, we adopt the same parameter settings for the other baseline models.

## Overall performance

Tables 2 and 3 show the performance of our method using the Precision@k, Recall@k, and NDCG@k evaluation metrics on both datasets compare to the other baseline models under normal conditions and cold start, respectively, where the k value is set to $\{10, 20\}$. The experimental results for some of the baseline models are derived from the values given by SocialLGN.

From the overall experimental results, it can be seen that SocialGCNRI achieves the optimal performance metrics in 18 out of the 24 evaluated metrics, which demonstrate that SocialGCNRI possesses a favorable recommendation performance. In the experimental comparison of various baselines, it can be found that the performance metrics of SBPR in both standard and cold-start situations are much smaller than those of the other baseline models (DiffNet, NGCF, LightGCN, SocialLGN, DSL, CLDS), which illustrates that the graph neural network-based recommendation model is able to more accurately capture the user's personalization than the traditional MF model preferences. In the recommendation model based on graph neural networks, DSL uses contrastive learning to mitigate the noisy information in the network connections, but it does not consider the interaction between the corresponding nodes, which makes the cross-views biased with information. SocialLGN incorporates social information based on LightGCN, and designs a method to

**Table 2 Comparison of recommendation performance on normal datasets.** (where P is Precision, R is Recall, and N is NDCG).

| Dataset | Metric | SBPR | DiffNet | NGCF | LightGCN | SocialLGN | DSL | CLDS | SocialGCNRI |
|---------|--------|------|---------|------|----------|-----------|-----|------|-------------|
| LastFM | P@10 | 0.1398 | 0.1727 | 0.1766 | 0.1961 | 0.1972 | 0.1974 | 0.1779 | **0.2005** |
| | P@20 | 0.1011 | 0.1215 | 0.1269 | 0.1358 | 0.1368 | **0.1399** | 0.1388 | 0.1391 |
| | R@10 | 0.1442 | 0.1779 | 0.1796 | 0.2003 | 0.2026 | 0.1995 | 0.2030 | **0.2045** |
| | R@20 | 0.2071 | 0.2488 | 0.2576 | 0.2769 | 0.2794 | 0.2823 | 0.2834 | **0.2837** |
| | N@10 | 0.1749 | 0.2219 | 0.2287 | 0.2536 | 0.2566 | 0.2383 | 0.2593 | **0.2621** |
| | N@20 | 0.1978 | 0.2474 | 0.2563 | 0.2788 | 0.2883 | 0.2812 | 0.2868 | **0.2891** |
| Ciao | P@10 | 0.0179 | 0.0238 | 0.0228 | 0.0271 | 0.0276 | 0.0208 | 0.0287 | **0.0303** |
| | P@20 | 0.0141 | 0.0182 | 0.0179 | 0.0202 | 0.0205 | 0.0217 | 0.0217 | **0.0225** |
| | R@10 | 0.0259 | 0.0341 | 0.0343 | 0.0410 | 0.0430 | 0.0560 | 0.0413 | **0.0651** |
| | R@20 | 0.0412 | 0.0527 | 0.0531 | 0.0591 | 0.0618 | 0.0560 | 0.0621 | **0.0651** |
| | N@10 | 0.0266 | 0.0359 | 0.0359 | 0.0437 | 0.0441 | 0.0436 | 0.0436 | **0.0478** |
| | N@20 | 0.0307 | 0.0403 | 0.0407 | 0.0478 | 0.0486 | 0.0488 | 0.0488 | **0.0521** |

**Note:**
Values in bold represent the best performance observed across all models.

**Table 3 Comparison of recommendation performance on cold-start datasets.** (where P is Precision, R is Recall, and N is NDCG).

| Dataset | Metric | SBPR | DiffNet | NGCF | LightGCN | SocialLGN | DSL | CLDS | SocialGCNRI |
|---------|--------|------|---------|------|----------|-----------|-----|------|-------------|
| LastFM | P@10 | 0.0292 | 0.0417 | 0.0333 | 0.0417 | 0.0458 | 0.0553 | **0.0667** | 0.0645 |
| | P@20 | 0.0333 | 0.0271 | 0.0292 | 0.0313 | 0.0333 | 0.0374 | 0.0417 | **0.0438** |
| | R@10 | 0.1123 | 0.1713 | 0.1169 | 0.1727 | 0.1974 | 0.2273 | **0.2676** | 0.2561 |
| | R@20 | 0.2467 | 0.2407 | 0.2141 | 0.2416 | 0.2663 | 0.2936 | 0.2911 | **0.3140** |
| | N@10 | 0.0709 | 0.1107 | 0.1074 | 0.1374 | 0.1419 | 0.1749 | 0.1887 | **0.1972** |
| | N@20 | 0.1159 | 0.1309 | 0.1411 | 0.1560 | 0.1643 | 0.1813 | **0.1939** | 0.1864 |
| Ciao | P@10 | 0.0070 | 0.0104 | 0.0104 | 0.0131 | 0.0134 | 0.0137 | 0.0141 | **0.0147** |
| | P@20 | 0.0060 | 0.0081 | 0.0085 | 0.0096 | 0.0097 | 0.0100 | 0.0101 | **0.0104** |
| | R@10 | 0.0234 | 0.0339 | 0.0341 | 0.0429 | 0.0441 | 0.0356 | **0.0452** | 0.0448 |
| | R@20 | 0.0384 | 0.0539 | 0.0557 | 0.0616 | 0.0630 | 0.0567 | **0.0684** | 0.0655 |
| | N@10 | 0.0165 | 0.0248 | 0.0245 | 0.0319 | 0.0328 | 0.0241 | 0.0343 | **0.0345** |
| | N@20 | 0.0219 | 0.0316 | 0.0319 | 0.0384 | 0.0394 | 0.0309 | 0.0417 | **0.0419** |

**Note:**
Values in bold represent the best performance observed across all models.

fuse user interaction information and social information, but it ignores the link between friendships and preferences, which makes SocialLGN underperform in cold starts although it achieves good performance in normal datasets. CLDS combines comparative learning and social information to construct the core information of social networks, and uses a heterogeneous graph neural network aggregation method to reduce the noisy data in the user interaction network, this method achieves good recommendation performance in different network environments, but the complex network architecture makes CLDS increase time loss when facing large data or complex networks. In contrast, SocialGCNRI significantly enhances information quality through the application of the FFT algorithms. Moreover, the model designs an adaptive residual graph convolution mechanism, which is

**Table 4 SocialGCNRI ablation experimental results on normal datasets.**

| Dataset | Metric | SocialIGNRI | Variant-P | Variant-G | Variant-F |
|---|---|---|---|---|---|
| LastFM | Precision@10 | **0.2005** | 0.1457 | 0.1957 | 0.2002 |
| | Precision@20 | **0.1391** | 0.1081 | 0.1387 | 0.1390 |
| | Recall@10 | **0.2069** | 0.1500 | 0.2002 | 0.2045 |
| | Recall@20 | **0.2837** | 0.2204 | 0.2381 | 0.2832 |
| | NDCG@10 | **0.2643** | 0.1845 | 0.2546 | 0.2641 |
| | NDCG@20 | **0.2891** | 0.2087 | 0.2848 | 0.2882 |
| Ciao | Precision@10 | **0.0303** | 0.0170 | 0.0270 | 0.0299 |
| | Precision@20 | **0.2252** | 0.0142 | 0.0207 | 0.2225 |
| | Recall@10 | **0.0442** | 0.0209 | 0.0408 | 0.0440 |
| | Recall@20 | **0.0651** | 0.0466 | 0.0620 | 0.0650 |
| | NDCG@10 | **0.0478** | 0.0242 | 0.0432 | 0.0472 |
| | NDCG@20 | **0.0521** | 0.0332 | 0.0488 | 0.0520 |

Note:
Values in bold represent optimal performance.

**Table 5 SocialGCNRI ablation experimental results on cold-start datasets.**

| Dataset | Metric | SocialGCNRI | Variant-P | Variant-G | Variant-F |
|---|---|---|---|---|---|
| LastFM | Precision@10 | **0.0645** | 0.0416 | 0.0500 | 0.0458 |
| | Precision@20 | **0.0438** | 0.2292 | 0.0375 | 0.0333 |
| | Recall@10 | **0.2561** | 0.1745 | 0.2143 | 0.1974 |
| | Recall@20 | **0.3140** | 0.1847 | 0.2709 | 0.2663 |
| | NDCG@10 | **0.1697** | 0.1310 | 0.1647 | 0.1419 |
| | NDCG@20 | **0.1864** | 0.1280 | 0.1855 | 0.1643 |
| Ciao | Precision@10 | **0.0147** | 0.0081 | 0.0131 | 0.0134 |
| | Precision@20 | **0.0104** | 0.0072 | 0.0099 | 0.0097 |
| | Recall@10 | **0.0448** | 0.0264 | 0.0428 | 0.0441 |
| | Recall@20 | **0.0655** | 0.0483 | 0.0631 | 0.0630 |
| | NDCG@10 | **0.0345** | 0.0189 | 0.0318 | 0.0328 |
| | NDCG@20 | **0.0419** | 0.0282 | 0.0394 | 0.0394 |

Note:
Values in bold represent optimal performance.

integrated into a heterogeneous GNN to deeply mine and analyze the complex interactions among users. In different network environments, SocialGCNRI has shown excellent recommendation performance, which fully proves its wide applicability and high efficiency in the real world application scenarios.

## Ablation study and analyses

The core innovative components of the SocialGCNRI model are two:

(1) The use of FFT and filter to process the original data for denoising.
(2) The proposal of an adaptive residual graph convolutional network to capture higher-order connections between users and items in the graph.
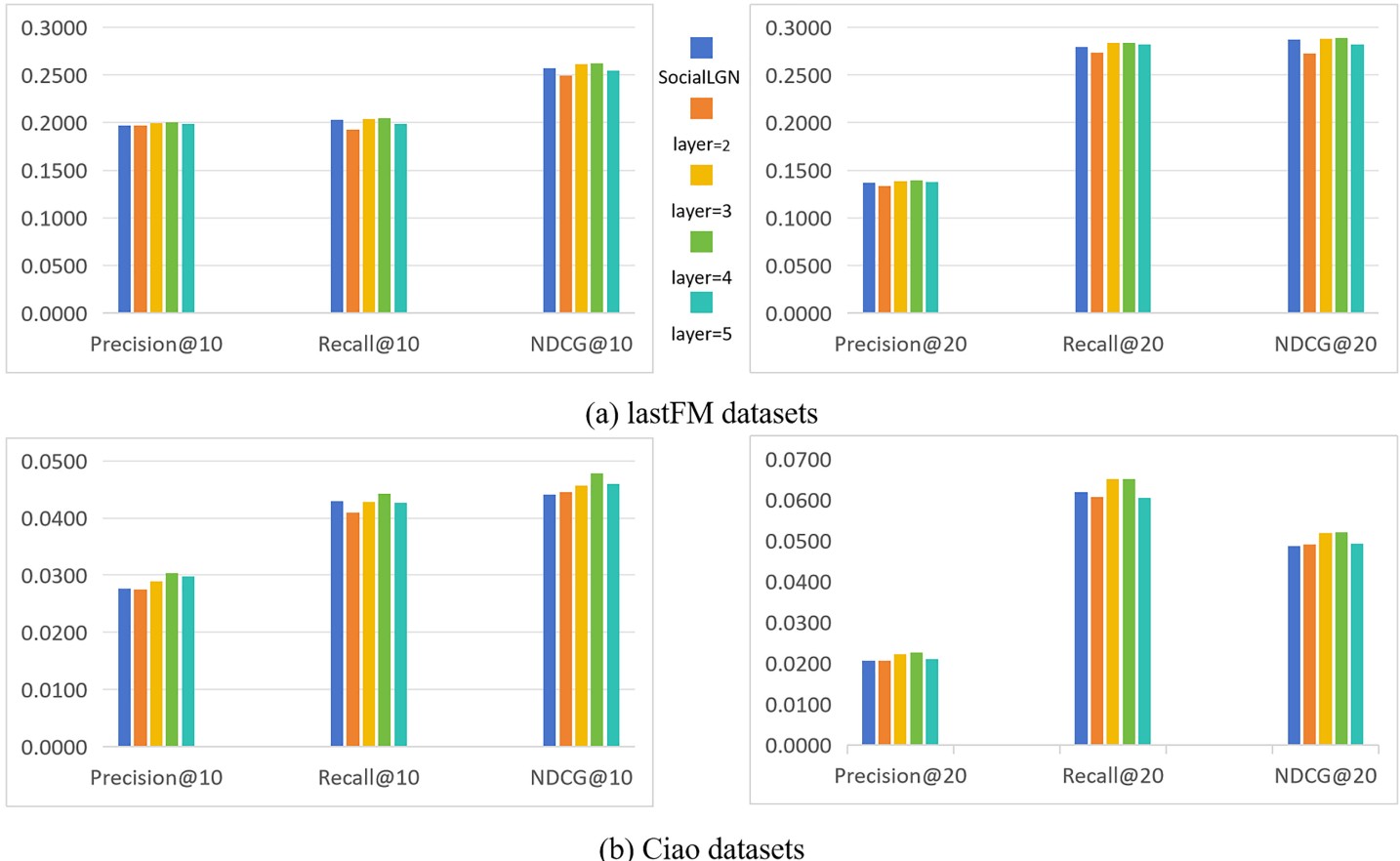

(a) lastFM datasets

(b) Ciao datasets

**Figure 2** (A and B) Impact of the number of convolution layers on recommendation performance (in normal datasets).

To justify these components, we set up three variants for comparison to show the impact of each component on the model. Variant-F denotes the SocialGCNRI model with filtering operations such as FFT removed, Variant-G denotes the SocialGCNRI model with adaptive residual graph convolution operations removed, and Variant-P denotes the SocialGCNRI model after removing the two core components. Tables 4 and 5 show the experimental results of SocialGCNRI and its three variants on the normal and cold-start datasets, respectively (the bold font indicates optimal performance).

We can see that all the performance metrics of the Variant-G model are improved to a certain extent compared with Variant-P, indicating that the user social relationship and item association relationship can be used to improve the recommendation performance by removing the noisy data. In addition, the Variant-F model, which includes an adaptive residual graph convolution algorithm, also exhibits a great improvement in performance compared to the Variant-P model, which shows that adaptive residual graph convolution algorithm can capture the higher-order connections between users and items more effectively. However, there is no significant difference between Variant-F and Variant-G in terms of recommendation performance, which indicates that the two core components

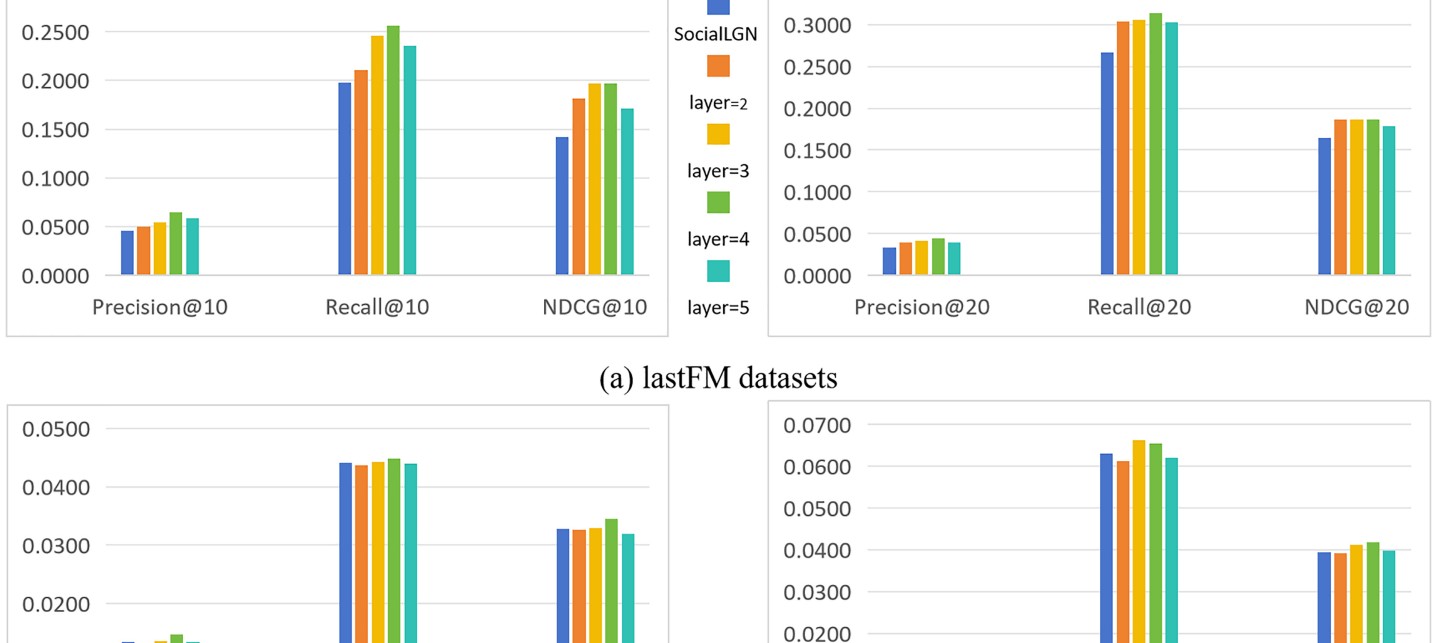

(a) lastFM datasets

(b) Ciao datasets

**Figure 3** (A and B) Impact of the number of convolution layers on recommendation performance (in cold-start datasets).

have the same importance for recommendation performance improvement. SocialGCNRI employs FFT to filter out the noisy data of social relations and user interactions, and uses an adaptive residual graph convolutional algorithm to further extract the effective information of the nodes in the graph and accurately capture the user's preferences, which demonstrates the optimal performance in comparison experiments on both datasets.

## Imapct of the number of convolutional layers

In this section, we take SocialLGN as a control (which uses LightGCN algorithm for the graph convolutional layers of the model), and analyze the effect of different number of convolutional layers in SocialGCNRI on the performance of the model by using Precision@k, Recall@k and NDCG@k as the evaluation metrics. Figure 2 shows the experimental results under the normal datasets, and Fig. 3 shows the cold start experimental results under the datasets.

As can be seen in Figs. 2 and 3, the performance of the SocialGCNRI model rises solidly as the number of convolutional layers increases. It can be further found that the

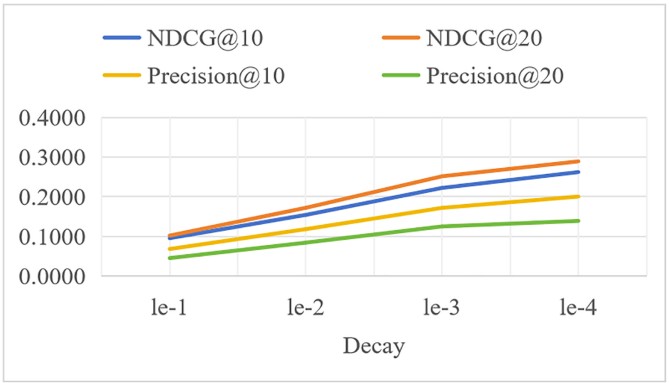 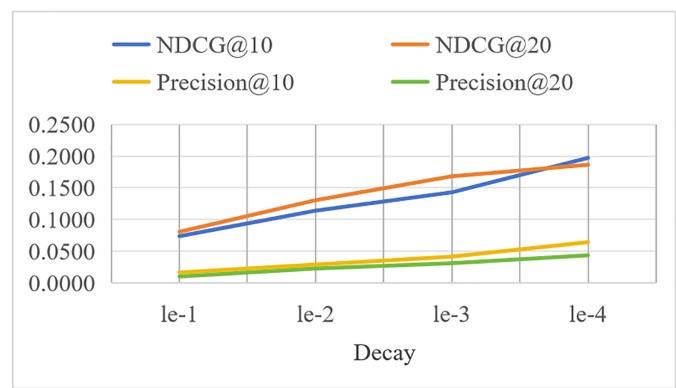

(a) Impact of weight decay on recommendation performance (experimental results on cold-start datasets on the right)

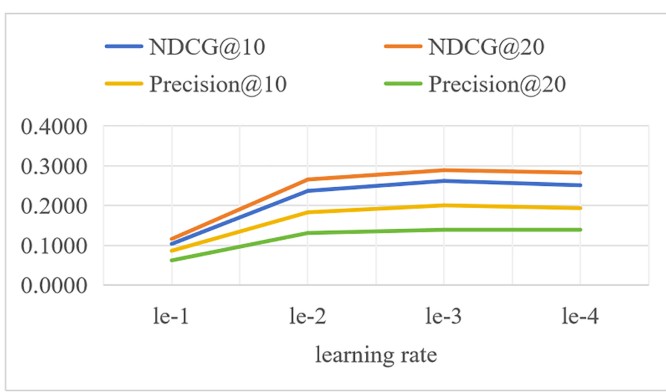 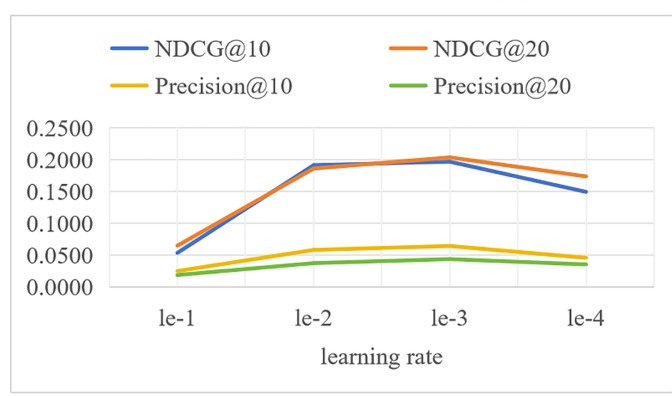

(b) Impact of learning rate on recommendation performance (experimental results on cold-start datasets on the right)

**Figure 4 (A and B) Impact of hyperparameters on recommendation performance on the lastfm datasets.**

SocialGCNRI reaches its optimal performance at layers = 4 and is much larger than at layers = 2, which indicates that our proposed adaptive residual graph convolution algorithm effectively mitigates the graph smoothing phenomenon and deepens the number of convolutional layers. The same can be seen in the comparison with SocialLGN, the performance of the two graph convolution algorithms is basically equal when the number of layers = 2. When the number of layers = 4, the overall effect of the adaptive residual graph convolution algorithm is much better than that of LightGCN, and this discrepancy is even more significant in the cold-start datasets. We think the possible reason for this is that, the increase in adaptive residual connectivity allows the nodes in the graph to always incorporate a certain amount of their own information during the convolution operation, which greatly slows down the emergence of the smoothing phenomenon in the graph, and makes the model more effective in capturing the deeper connections between the nodes.

## Impact of hyperparameters on recommendation performance

During the training process of the model, different hyperparameter settings can directly affect the performance of the model. We evaluate the impact of key hyperparameters such

as learning rate (lr) and weight decay coefficient (decay) on the recommendation performance of the SocialGCNRI model in this section. The experimental results are shown in Fig. 4.

The weight decay coefficient is an important parameter to regulate the strength of the $L_2$ regularization penalty. As this coefficient increases, the penalty for larger weights increases, driving the model weights towards smaller values. However, over-increasing the coefficient may lead to underfitting of the model, meaning that the model is too simplified to capture the complex structure behind the data. Conversely, if the weight decay coefficients are too small, the weights are not penalized enough and the model may be at risk of overfitting. Therefore, selecting an appropriate weight decay coefficient is crucial for the generalization ability of the model, which is directly related to the performance of the model. In this section of experiments, we set the size of the weight decay coefficient coefficients to {1e−1, 1e−2, 1e−3, 1e−4, 1e−5}, and analyze the effect of different sizes of weight decay coefficients on the model performance through the experiments conducted on Lastfm datasets. From the experimental results shown in Fig. 4, it can be seen that the SocialGCNRI model exhibits the best performance when the weight decay coefficient is set to 1e−4.

The learning rate refers to the magnitude of the model weight update during the optimization process, and the difference in the size of the learning rate value will directly affect the final performance of the model. Too small a learning rate makes the training process slow, which may cause the model to fall into local minima and reach only sub-optimal solutions. While a higher learning rate allows for fast convergence, the training process may be unstable and prone to overfitting. In our experiments, we set the size of the learning rate to {1e−1, 1e−2, 1e−3, 1e−4} and analyze the effect of different learning rate sizes on the model performance. From the experimental results shown in Fig. 4, it can be seen that the model exhibits the best performance when the learning rate is 1e−3.

## CONCLUSION

In this article, we propose a social recommendation model based on adaptive residual graph convolutional network (SocialGCNRI). We mine the friend relationships between users and the correlations between items from the user-item interaction matrix, and fuse this information into a heterogeneous graph to alleviate the problem of data sparsity. In addition, the FFT algorithm is used to remove the noise data in the heterogeneous graph and user social information. Subsequently, we design an adaptive residual graph convolution algorithm to accurately extract user and item representations in both views. Finally, the information fusion is performed by MLP to achieve accurate personalized recommendation. A series of experiments on two publicly available datasets demonstrate that SocialGCNRI achieves excellent recommendation performance in different situations.

In future research, we will further focus on the noisy data in the model. For example, by incorporating timestamps for more accurate denoising, and we also consider the use of

**Table 6 Statistics of the Douban datasets.**

| Datasets | Number of users | Number of item | Number of user-item interaction | Density of interaction | Number of social connections | Density of connections |
|---|---|---|---|---|---|---|
| Douban | 2,848 | 39,586 | 894,887 | 0.793% | 35,770 | 0.441% |

**Table 7 Comparison of recommendation performance on normal datasets.**

| Dataset | Metric | LightGCN | SocialLGN | DSL | SocialGCNRI |
|---|---|---|---|---|---|
| Douban | Precision@10 | 0.1398 | 0.1727 | 0.1766 | **0.1961** |
| | Precision@20 | 0.2238 | 0.2345 | 0.2365 | **0.2821** |
| | Recall@10 | 0.1965 | 0.2035 | 0.2054 | **0.2213** |
| | Recall@20 | 0.0508 | 0.0628 | 0.0524 | **0.0625** |
| | NDCG@10 | 0.2437 | 0.2579 | 0.2335 | **0.2783** |
| | NDCG@20 | 0.2269 | 0.2405 | 0.2275 | **0.2571** |

Note:
Values in bold represent optimal performance.

**Table 8 Comparison of recommendation performance on cold-start datasets.**

| Dataset | Metric | LightGCN | SocialLGN | DSL | SocialGCNRI |
|---|---|---|---|---|---|
| Douban | Precision@10 | 0.0162 | 0.0261 | 0.0212 | **0.0262** |
| | Precision@20 | 0.0131 | 0.0205 | 0.0193 | **0.0213** |
| | Recall@10 | 0.0722 | 0.1073 | 0.0854 | **0.1131** |
| | Recall@20 | 0.1128 | 0.1672 | 0.1380 | **0.1659** |
| | NDCG@10 | 0.0464 | 0.0608 | 0.0556 | **0.0763** |
| | NDCG@20 | 0.0594 | 0.0908 | 0.0720 | **0.0921** |

Note:
Values in bold represent optimal performance.

contrastive learning to create more perspectives for the user to further improve the quality of node embedding and recommendation performance.

## APPENDIX

To further demonstrate the generalizability of the models, we selected LightGCN, SocialLGN and DSL as the baseline models, and conducted comparative experiments with SocialGCNRI on the douban datasets. The dataset statistics are shown in Table 6.

Tables 7 and 8 show the performance of our method using the Precision@k, Recall@k, and NDCG@k evaluation metrics on douban datasets compare to the baseline models under normal conditions and cold start, respectively, where the k value is set to {10, 20}.

According to the performance metrics on the normal Douban datasets, the SocialGCNRI model improves on Precision@10/20 by 11.04% and 19.28%, respectively, compared to optimal baseline model. Similarly, the SocialGCNRI model improves 7.74% and 19.27% on Recall@10/20, and 19.18, and 13.01% on NDCG@10/20, respectively. Even when tested on the cold-start datasets, the SocialGCNRI model significantly outperforms other baseline models.

## Funding

This work was supported by the Major Science and Technology Programs in Henan Province under Grants 241100210100, by the National Natural Science Foundation of China under Grants 62102372, 62072416, and 61902361, by Key Research and Development Program of Shaanxi (Program No. 2024GX-YBXM-545), by Key Research and Development Special Project of Henan Province under Grants 252102211070, 252102210139, 252102210127, 232102211051, 232102211053, 242102211068, 242102210107, 232102321069, 232102210078, HNKP2024214, and 2023SJGLX369Y, by the Natural Science Foundation Project of Henan Province under Grant 222300420582, by the Doctoral Fund Project of Zhengzhou University of Light Industry under Grants 2021BSJJ029, and 2020BSJJ030, by the Mass Innovation Space Incubation Project under Grant 2023ZCKJ216, by the Key Scientific Research Projects of Higher Education Institutions in Henan Province under Grant 24B520038, and by the innovation team of data science and knowledge engineering of Zhengzhou University of Light Industry. The funders had no role in study design, data collection and analysis, decision to publish, or preparation of the manuscript.

## Grant Disclosures

The following grant information was disclosed by the authors:
Major Science and Technology Programs in Henan Province: 241100210100.
National Natural Science Foundation of China: 62102372, 62072416, and 61902361.
Key Research and Development Program of Shaanxi: 2024GX-YBXM-545.
Key Research and Development Special Project of Henan Province: 252102211070, 252102210139, 252102210127, 232102211051, 232102211053, 242102211068, 242102210107, 232102321069, 232102210078, HNKP2024214, and 2023SJGLX369Y.
Natural Science Foundation Project of Henan Province: 222300420582.
Doctoral Fund Project of Zhengzhou University of Light Industry: 2021BSJJ029, and 2020BSJJ030.
Mass Innovation Space Incubation Project: 2023ZCKJ216.
Key Scientific Research Projects of Higher Education Institutions in Henan Province: 24B520038.
Innovation Team of Data Science and Knowledge Engineering of Zhengzhou University of Light Industry.

## Competing Interests

The authors declare that they have no competing interests.

## Author Contributions

- Rui Chen conceived and designed the experiments, performed the experiments, performed the computation work, prepared figures and/or tables, authored or reviewed drafts of the article, and approved the final draft.

- Kangning Pang conceived and designed the experiments, performed the experiments, performed the computation work, prepared figures and/or tables, authored or reviewed drafts of the article, and approved the final draft.
- Qingfang Liu performed the experiments, analyzed the data, authored or reviewed drafts of the article, and approved the final draft.
- Lei Zhang performed the experiments, analyzed the data, prepared figures and/or tables, and approved the final draft.
- Hao Wu analyzed the data, authored or reviewed drafts of the article, and approved the final draft.
- Cundong Tang performed the computation work, prepared figures and/or tables, authored or reviewed drafts of the article, and approved the final draft.
- Pu Li analyzed the data, prepared figures and/or tables, and approved the final draft.

## Data Availability

The code is available in the Supplemental File.

The data used in this study is publicly available (https://grouplens.org/datasets/hetrec-2011/, https://www.cse.msu.edu/tangjili/datasetcode/truststudy.html, https://github.com/WorkingMiracles/dataset.git).

## Supplemental Information

Supplemental information for this article can be found online at http://dx.doi.org/10.7717/peerj-cs.3010#supplemental-information.

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
