# Peer review of "A social recommendation model based on adaptive residual graph convolution networks"

_PeerJ Computer Science, doi:10.7717/peerj-cs.3010_

## Round 0.1 · original submission · Minor Revisions

**Language Note:** The review process has identified that the English language must be improved. PeerJ can provide language editing services - please contact us at [email protected] for pricing (be sure to provide your manuscript number and title). Alternatively, you should make your own arrangements to improve the language quality and provide details in your response letter. – PeerJ Staff

Reviewer 1 ·

Basic reporting

Summary:
The paper “A Social Recommendation Model Based on Adaptive Residual Graph Convolution Networks (SocialGCNRI)” proposes a novel approach for social recommendation that leverages graph neural networks (GNNs) to address data sparsity, cold-start issues, and noise in raw data. The authors introduce a model that combines the Fast Fourier Transform (FFT) to filter noisy data in the frequency domain and an adaptive residual graph convolutional network to capture higher-order connections between nodes in heterogeneous graphs. The model shows superior performance compared to several state-of-the-art methods, as demonstrated by experiments on two real-world datasets (LastFM and Ciao). The results highlight the model’s robustness in both standard and cold-start scenarios, with significant improvements in metrics such as Precision@K, Recall@K, and NDCG@K.

Strengths:
1. Novel Integration of FFT for Noise Reduction
The use of FFT as a filtering mechanism to reduce noise in user-item interaction data is a unique contribution to the recommendation domain. This technique effectively addresses noisy data caused by user misoperations and improves the quality of input embeddings.
2. Adaptive Residual Graph Convolution
The design of an adaptive residual mechanism for graph convolution networks is well-motivated. By adaptively incorporating initial embedding information and delaying the graph smoothing issue, the model effectively captures higher-order connections in heterogeneous graphs, leading to improved recommendation accuracy.
3. Comprehensive Experiments and Ablation Studies
The paper provides extensive experimental results, comparing SocialGCNRI with multiple baseline models across both normal and cold-start datasets. The inclusion of ablation studies (removing FFT and residual graph convolution) clearly demonstrates the contributions of the proposed innovations to the model’s performance.


Weaknesses
1. Limited Discussion on Cold-Start Recommendation Models
While the paper addresses the cold-start problem and shows performance improvements, it lacks sufficient discussion of existing cold-start recommendation models and methods. Some related works are missing from the reference[1-4]. These models directly tackle the cold-start issue and should be reviewed to provide context for the proposed approach’s contributions and limitations. Suggested Action: Include a comparative discussion of how SocialGCNRI differs from or complements these methods in handling cold-start scenarios.
2. Insufficient Discussion on Debiasing Methods
The paper does not adequately address biases commonly present in recommendation systems, such as popularity bias or interaction bias. Many recent works focus on debiasing models to improve fairness and accuracy in recommendations. Missing citations[5-6] and discussions include of debiasing recommendation. These methods provide insights into removing biases in user-item interactions, which could be a relevant enhancement for SocialGCNRI. Discuss how SocialGCNRI handles or could handle such biases, particularly in noisy environments.
3. Clarity and Overemphasis on Heterogeneous Graphs
While the focus on heterogeneous graphs is compelling, the paper does not thoroughly discuss the trade-offs of using such a complex architecture. For instance, scalability issues with large graphs or computational overheads introduced by the residual connections and FFT are not analyzed. Further, while the authors evaluate performance on two datasets, additional diverse datasets (e.g., e-commerce or academic recommendation datasets) could better validate the model’s generalizability.

[1] Generative adversarial framework for cold-start item recommendation
[2] Aligning Distillation For Cold-start Item Recommendation
[3] Large Language Model Simulator for Cold-Start Recommendation
[4] Cold-Start Recommendation towards the Era of Large Language Models (LLMs): A Comprehensive Survey and Roadmap
[5] Adaptive Popularity Debiasing Aggregator for Graph Collaborative Filtering
[6] Bias and Debias in Recommender System: A Survey and Future Directions

Experimental design

Comprehensive Experiments and Ablation Studies
The paper provides extensive experimental results, comparing SocialGCNRI with multiple baseline models across both normal and cold-start datasets. The inclusion of ablation studies (removing FFT and residual graph convolution) clearly demonstrates the contributions of the proposed innovations to the model’s performance.

Validity of the findings

Novel Integration of FFT for Noise Reduction
The use of FFT as a filtering mechanism to reduce noise in user-item interaction data is a unique contribution to the recommendation domain. This technique effectively addresses noisy data caused by user misoperations and improves the quality of input embeddings.

Reviewer 2 ·

Basic reporting

The paper proposes an innovative social recommendation model, and while I am from a different domain (decentralized systems and reputation-based consensus) I was struck by strong conceptual parallels between the two fields. I encourage the authors to consider my comments in that context, though I leave domain-specific evaluation to experts in recommender systems.

Experimental design

no comment

Validity of the findings

no comment

Additional comments

1. The manuscript would benefit from explicitly framing social recommendation via trust graphs as conceptually aligned with reputation-based consensus in decentralized systems (e.g., blockchain). The propagation of user influence in the model mirrors how trust and reputation are aggregated in systems like identity-based Proof-of-Stake.

2. The authors should highlight that noise or manipulation in social interactions is analogous to Sybil attacks in distributed systems. Techniques like FFT filtering and trust-based weighting correspond closely to anti-Sybil mechanisms in consensus protocols. Strengthening this analogy would deepen the reader’s understanding.

3. Please consider discussing how your model implicitly balances openness with trustworthiness—a well-known challenge in decentralized networks.

Reviewer 3 ·

Basic reporting

This paper proposes a novel social recommendation model called SocialGCNRI, which leverages the Fast Fourier Transform to reduce the impact of noise on recommendation performance. The structure of the paper is complete and the logic is relatively clear. The authors conduct comparative experiments, ablation studies, and analyses on two datasets to verify the effectiveness of the proposed method.

The expression in the paper can be further improved. For example, constants, variables, and matrices should be represented in distinct styles; additionally, the model diagram should be arranged horizontally to enhance readability.

Experimental design

The paper conducts comparative experiments on two datasets. However, it lacks the hyperparameter settings for all the baseline methods. In addition, comparing with more recent social recommendation methods—especially noise-resistant social recommendation models—could further demonstrate the effectiveness of the proposed approach.

Validity of the findings

The proposed method demonstrates a certain degree of novelty, and the introduction of the method is sufficient and clear. Conducting comparative experiments with more methods of a similar type could further strengthen the validity of the paper’s conclusions.

---

## Round 0.2 · accepted · Accept

Based on the reviewers' reports, and my own assessment as Editor, I am pleased to inform you that the manuscript is acceptable for publication in PeerJ Computer Science.

Reviewer 2 ·

Basic reporting

My earlier comments were all addressed

Experimental design

My earlier comments were all addressed

Validity of the findings

My earlier comments were all addressed

Additional comments

My earlier comments were all addressed

Reviewer 3 ·

Basic reporting

This paper proposes a novel social recommendation model called SocialGCNRI, which leverages the Fast Fourier Transform to reduce the impact of noise on recommendation performance. The structure of the paper is complete and the logic is relatively clear. The authors conduct comparative experiments, ablation studies, and analyses on two datasets to verify the effectiveness of the proposed method.

Experimental design

The paper conducts comparative experiments on two datasets. In the revised manuscript, the authors have added new comparative methods and new datasets, which enhance the validity of the paper's conclusions.

Validity of the findings

The proposed method demonstrates a certain degree of novelty, and the introduction of the method is sufficient and clear. Conducting comparative experiments with more methods of a similar type could further strengthen the validity of the paper’s conclusions.

Additional comments

The authors' response has addressed my concerns, and I have no further questions.